# COVID-19 Survivor Patients Carrying the Rs35705950 Risk Allele in *MUC5B* Have Higher Plasma Levels of Mucin 5B

Salvador García-Carmona [1], Ramcés Falfán-Valencia [1], Abigail Verónica-Aguilar [1], Ivette Buendía-Roldán [2], Leslie Chávez-Galán [3], Rafael de Jesús Hernández-Zenteno [4], Alfonso Martínez-Morales [1], Ingrid Fricke-Galindo [1], Jesús Alanis-Ponce [1], Daniela Valencia-Pérez Rea [1], Ilse Adriana Gutiérrez-Pérez [1,5], Oscar Zaragoza-García [5], Karol J. Nava-Quiroz [1], Angel Camarena [1], Mayra Mejía [6], Iris Paola Guzmán-Guzmán [5] and Gloria Pérez-Rubio [1,*]

1   HLA Laboratory, Instituto Nacional de Enfermedades Respiratorias Ismael Cosío Villegas, Mexico City 14080, Mexico; q.c.salvadorgc@gmail.com (S.G.-C.); rfalfanv@iner.gob.mx (R.F.-V.); 14602108@uagro.mx (A.V.-A.); 17296572@uagro.mx (A.M.-M.); ifricke@iner.gob.mx (I.F.-G.); 14alanisponce@gmail.com (J.A.-P.); daniela.980211@gmail.com (D.V.-P.R.); ilse.gutierrez@iner.gob.mx (I.A.G.-P.); knava@iner.gob.mx (K.J.N.-Q.); ang_edco@yahoo.com.mx (A.C.)

2   Translational Research Laboratory on Aging and Pulmonary Fibrosis, Instituto Nacional de Enfermedades Respiratorias Ismael Cosío Villegas, Mexico City 14080, Mexico; ivettebu@yahoo.com.mx

3   Laboratory of Integrative Immunology, Instituto Nacional de Enfermedades Respiratorias Ismael Cosío Villegas, Mexico City 14080, Mexico; lchavezgalan@gmail.com

4   COPD Clinic, Instituto Nacional de Enfermedades Respiratorias Ismael Cosío Villegas, Mexico City 14080, Mexico; rafherzen@yahoo.com.mx

5   Faculty of Chemical-Biological Sciences, Universidad Autónoma de Guerrero, Chilpancingo 39087, Mexico; zaragoza789@hotmail.com (O.Z.-G.); pao_nkiller@yahoo.com.mx (I.P.G.-G.)

6   Interstitial Pulmonary Diseases and Rheumatology Unit, Instituto Nacional de Enfermedades Respiratorias Ismael Cosío Villegas, Mexico City 14080, Mexico; medithmejia1965@gmail.com

\*   Correspondence: gperezrubio@iner.gob.mx; Tel.: +52-55-5487-1700 (ext. 5152)

**Abstract:** Background: Genetic susceptibility to infectious diseases is partly due to the variation in the human genome, and COVID-19 is not the exception. This study aimed to identify whether risk alleles of known genes linked with emphysema (*SERPINA1*) and pulmonary fibrosis (*MUC5B*) are associated with severe COVID-19, and whether plasma mucin 5B differs according to patients' outcomes. Materials and methods: We included 1258 Mexican subjects diagnosed with COVID-19. We genotyped rs2892474 and rs17580 of the *SERPINA1* gene and rs35705950 of *MUC5B*. Based on the rs35705950 genotypes, mucin 5B plasma protein levels were quantified. Results: Homozygous for the risk alleles of the three polymorphisms were found in less than 5% of the study population, but no statistically significant difference in the genotype or allele association analysis. At the protein level, non-survivors carrying one or two copies of the risk allele rs35705950 in *MUC5B* (GT + TT) had lower levels of mucin 5B compared to the survivors (0.0 vs. 0.17 ng/mL, $p = 0.0013$). Conclusion: The polymorphisms rs28929474 and rs17580 of *SERPINA1* and rs35705950 of *MUC5B* are not associated with the risk of severe COVID-19 in the Mexican population. COVID-19 survivor patients bearing one or two copies of the rs35705950 risk allele have higher plasma levels of mucin 5B.

**Keywords:** *MUC5B*; COVID-19; rs35705950; A1AT; *SERPINA1*; mucin 5B

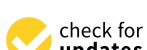



## 1. Introduction

Coronavirus disease 2019 (COVID-19) is caused by SARS-CoV-2, a virus from the coronavirus family; after infecting the human, this can be asymptomatic or present as a common cold to more severe forms of the disease [1,2]; about 13% of the infected develop severe symptoms [3].

Genetic susceptibility to disease or infection is partly due to the variation in the human genome, and COVID-19 is no exception. Genome-wide association studies (GWASs) have

identified variants [4] that correspond to genes whose transcripts participate in the immune response or fulfill crucial functions at the pulmonary level [5]; these have been associated with the susceptibility and progression of the disease [4]. Among the candidate genes to be evaluated are *SERPINA1*, which codes for the alpha 1 antitrypsin protein (A1AT) [6], and *MUC5B*, which codes for mucin 5B [7]. Reduced plasma levels of the protein characterize A1AT deficiency, which is caused by polymorphisms in *SERPINA1*. The most relevant, PiS and PiZ (allele T [rs17580] and allele A [rs2892474], respectively), have been associated with increased risks of pulmonary emphysema and chronic obstructive pulmonary disease (COPD) [8]. The polymorphism in *MUC5B* (rs35705059) is located approximately 3 kb before the transcription start-site, achieving an enhancer function and affecting protein expression. The T allele has been associated with the risk of idiopathic pulmonary fibrosis (IPF) since these patients' mucin 5B levels are elevated [7].

Both genes have been significantly associated with lung diseases such as COPD and IPF. The presence of the risk alleles affects the plasmatic concentrations of proteins of the immune response; additionally, studies regarding the genetic susceptibility of suffering from COVID-19 and these genetic polymorphisms are scanty. This study aimed to identify whether the presence of alleles A (rs2892474) and T (rs17580) of the *SERPINA1* gene, and allele T (rs35705950) of *MUC5B* in patients with COVID-19 are associated with severe forms of the disease in the Mexican population, and to evaluate the concentration of mucin 5B in the plasma of patients with severe COVID-19.

## 2. Materials and Methods

### 2.1. Selection of the Study Population

We included 1258 Mexican patients hospitalized in the Instituto Nacional de Enfermedades Respiratorias Ismael Cosío Villegas (Mexico City, Mexico) diagnosed with COVID-19. Only patients with a positive SARS-CoV-2 RT-PCR test and $\geq$18 years old were consecutively enrolled and signed informed consent. The study protocol was approved by the local Research Ethics Committee (C53-20) and complied with the Helsinki Declaration statements.

Three comparisons were made. The first included those patients who required invasive mechanical ventilation (IMV) and were compared with patients who did not require it (non-IMV). For the second analysis, they were divided into three groups: mild COVID-19 (mild)—those patients between 200 and 300 mmHg $PaO_2/FiO_2$ at admission; moderate COVID-19 (moderate)—between 100 to 200 mmHg $PaO_2/FiO_2$; and severe COVID-19 (severe) for those with <100 mmHg $PaO_2/FiO_2$. Finally, a comparison was performed according to the patients' outcomes, classified into survivors and non-survivors.

### 2.2. DNA Extraction and Quantification

Peripheral blood was obtained from each patient in tubes with EDTA as an anticoagulant, with prior informed consent. The genetic material was extracted using the BDtractTM kit (Maxim Biotech Inc., San Francisco, CA, USA). The DNA obtained was quantified in a Nanodrop 2000 (Thermo Scientific, Wilmington, DE, USA). The ratio 260/280 was obtained to evaluate the purity of the sample, and samples were considered free of contaminants with a ratio of 1.8–2.0.

### 2.3. Genotyping

The alleles and genotypes of each polymorphism were identified by real-time PCR using the StepOnePlus device (Applied Biosystems, Foster City, CA, USA) by allelic discrimination through predesigned TaqMan probes as follows: rs2892474 (C__27295323_10), rs17580 (C____594695_20) of the *SERPINA1*, and rs35705950 (C___1582254_20) of *MUC5B* (Applied Biosystems, Foster City, CA, USA). Four negative controls were included per plate, and 1% of the samples included in the study were genotyped in duplicate as allele assignment control.

## 2.4. Plasma Mucin 5B Measurement

Mucin 5B quantification was performed in plasma using an ELISA assay (MUC5B ELISA kit Human Mucin 5 Subtype B (MUC5B) ELISA Kit, My BioSource, catalogue number MBS2024599, San Diego, CA, USA), following the manufacturer's instructions. A subgroup of 144 patients was selected according to the rs35705950 (*MUC5B*) genotype consisting of 57 GG patients, 84 heterozygotes (GT), and 3 TT patients. The comparison was performed according to the patients' outcomes, classified into survivors and non-survivors.

## 2.5. Statistical Analysis

We used Epidat version 3.1 (Xunta de Galicia and Pan American Health Organization, 2006) and SPSS version 15.0 (IBM Corp. Released, 2011) to describe the demographic and clinical variables of the study population. The Kolmogorov–Smirnov test was performed to evaluate the distributions of the quantitative variables, and we applied the Mann–Whitney-U test because the variables' distributions were non-normal. The Hardy–Weinberg equilibrium (HWE) was calculated for the three polymorphisms. The association by genotypes and alleles was performed using Epidat version 3.1 (Xunta de Galicia and Pan-American Health Organization, 2006); genetic models were applied. Linkage disequilibrium (LD) analysis for *SERPINA1* gene polymorphisms was performed in Haploview version 4.1 (Barrett et al., 2005). RStudio version 4.1.2 was used to analyze the MUC5B in plasma concentration. We employed the Mann–Whitney-U test to compare the protein levels in the study groups.

## 3. Results

### 3.1. Study Population

The non-survivors were older than the survivors (63 vs. 56 years, $p < 0.001$). A higher frequency of men was observed in the non-survivors group (71.9%). Regarding the clinical variables, the frequencies of fever (66.4 vs. 73.3, $p = 0.012$), anosmia (3.9 vs. 8.6, $p = 0.002$) and ageusia (9.3 vs. 14.8, $p = 0.006$) showed differences between non-survivors and survivors. It should be noted that non-survivors had more days of hospitalization (20 vs. 19, $p = 0.029$) and had lower $PaO_2/FiO_2$ at hospital admission compared to survivors (128 vs. 166, $p < 0.001$), and a high proportion of patients had a requirement for invasive mechanical ventilation (93.0 vs. 60.9, $p < 0.001$) (Table 1).

### 3.2. Genotype and Allele Association

The polymorphisms analyzed complied with the HWE ($p = 0.591$ for rs17580, $p = 0.943$ in rs28929474, and $p = 0.091$ for rs35705950). The rs17580 heterozygous AT genotype was more frequent in the survivors group when compared to the non-survivors group (3.91% vs. 3.53%); the rs28929474 heterozygous CT only was only present in the survivors (1.01%); and rs35705950 (GT) heterozygosity showed similar frequencies in both comparison groups. There were no statistically significant differences in the comparisons between genotypes. Homozygous risk alleles for the three polymorphisms were found in less than 5%, regardless of the study group, and there was no statistically significant difference (Table 2).

Regarding the allele frequency analysis, the minor allele of rs17580 (A) occurred more frequently in the survivors group (2.08%) compared to the non-survivors group (1.77%). The T allele of rs28929274 was present in the survivors group (0.50%), and the frequency of the risk allele (T) of rs35705950 was lower (3.5%) in survivors compared to those non-survivors (3.60%). There were no statistically significant differences at the allele level (Table 2).

According to the requirements of invasive mechanical ventilation (Supplementary Table S1) and ARDS (Supplementary Table S2), there were no statistically significant differences in genotypes or alleles when performing the genetic analysis.

**Table 1.** Demographic and clinical characteristics of the COVID-19 patients included in this study.

| Variables | Non-Survivors $n = 460$ | Survivors $n = 798$ | $p$ |
|---|---|---|---|
| Age (years) | 63 (48–64) | 56 (48–64) | **<0.001** |
| Male, $n$ (%) | 331 (71.9) | 509 (63.8) | **0.003** |
| BMI (kg/m$^2$) | 29 (26–33) | 30 (27–33) | **0.004** |
| Comorbidities % | | | |
| Obesity | 41.7 | 47.2 | 0.068 |
| T2DM | 31.1 | 27.1 | 0.150 |
| CRD | 9.3 | 5.9 | **0.029** |
| IHC | 6.3 | 3.0 | **0.007** |
| SAH | 38.2 | 33.7 | 0.122 |
| Onset of symptoms (days) | 9 (7–14) | 9 (7–12) | 0.574 |
| Symptoms % | | | |
| Fever | 66.4 | 73.3 | **0.012** |
| Cough | 68.2 | 67.4 | 0.836 |
| Dyspnea | 84.7 | 83.9 | 0.760 |
| Myalgia | 60.1 | 65.6 | 0.057 |
| Arthralgias | 58.9 | 62.3 | 0.251 |
| Headache | 42.7 | 45.9 | 0.295 |
| Rhinorrhea | 17.6 | 14.9 | 0.245 |
| Odynophagia | 22.4 | 26.1 | 0.570 |
| Diarrhea | 8.5 | 11.3 | 0.138 |
| Vomiting | 2.4 | 3.1 | 0.559 |
| Chest pain | 7.4 | 10.9 | 0.536 |
| Anosmia | 3.9 | 8.6 | **0.002** |
| Ageusia | 9.3 | 14.8 | **0.006** |
| Hospitalization (days) | 20 (13–30) | 19 (12–30) | **0.029** |
| mmHg PaO$_2$/FiO$_2$ | 128 (86–169) | 166 (114–212) | **<0.001** |
| IMV, $n$ (%) | 93.0 | 60.9 | **<0.001** |

BMI: body mass index, T2DM: type 2 diabetes mellitus, CRD: chronic respiratory disease, IHC: ischemic heart disease, SAH: systemic arterial hypertension, IMV: invasive mechanical ventilation. We show quantitative variables as median and 25th and 75th percentiles; $p$-value was evaluated using the Mann–Whitney U test; categorical variables are shown as percentages and were acquired using χ2 testing. Significant differences are highlighted in bold.

**Table 2.** Genotype and allele frequencies of *SERPINA1* and *MUC5B* polymorphisms in patients with COVID-19 discharged due to death or improvement.

| Genotypes/ Alleles | Non-Survivors $n = 458$ (%) | Survivors $n = 793$ (%) | $p$-Value | OR | CI, 95% |
|---|---|---|---|---|---|
| rs17580 | | | | | |
| TT | 96.47 | 95.96 | | 1 (Reference) | |
| TA | 3.53 | 3.91 | 0.587 | 0.89 | 0.48–1.65 |
| AA | 0 | 0.13 | | | |
| T | 98.23 | 97.92 | 0.690 | | |
| A | 1.77 | 2.08 | | 0.84 | 0.46–1.54 |
| rs28929474 | | | | | |
| CC | 99.74 | 98.99 | | 1 (Reference) | |
| CT | 0.13 | 1.01 | 0.550 | 0.21 | 0.02–1.72 |
| TT | 0.13 | 0 | | | |
| C | 99.74 | 99.50 | 0.213 | | |
| T | 0.26 | 0.50 | | 0.21 | 0.02–1.72 |
| rs35705950 | | | | | |
| GG | 93.01 | 93.57 | | 1 (Reference) | |
| GT | 6.77 | 6.18 | 0.735 | 1.10 | 0.69–1.75 |
| TT | 0.22 | 0.25 | | 0.87 | 0.07–9.63 |
| G | 96.39 | 96.66 | 0.817 | | |
| T | 3.60 | 3.34 | | 1.08 | 0.69–1.67 |

### 3.3. Linkage Disequilibrium

Linkage disequilibrium (LD) analysis for rs28929474 and rs17580 in the *SERPINA1* gene in the study population showed that these two genetic variants were not in LD in patients with severe COVID-19 survivors and non-survivors (Supplementary Figure S1).

### 3.4. Evaluation of Mucin 5B Plasma Levels

Patients with the three genotypes according to rs35705950 in *MUC5B* were selected. There was not a statistically significant difference between patients carrying the homozygous genotype for the common allele (GG) when compared to those carrying one or two copies of the risk allele (GG + TT) (Supplementary Figure S2). There was a statistically significant difference between those individuals carrying one or two copies of the risk allele (GT + TT) as analyzed by the outcome of each patient (Figure 1). Non-survivors had lower median levels of mucin 5B compared to the survivors (0.0 vs. 0.17 ng/mL, respectively); this difference was significant ($p = 0.0013$). A linear regression analysis was performed to assess whether the significant differences in mucin 5B concentrations depending on the outcome are not affected by possible confounding variables (age, sex, BMI). After this analysis, the significance *p*-value was preserved ($p = 0.0058$).

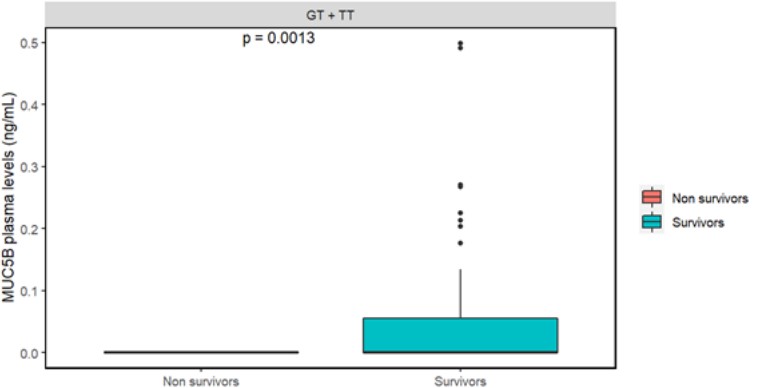

**Figure 1.** Concentrations of mucin 5B in plasma from patients with severe COVID-19. We employed the Mann–Whitney-U test to compare the protein level in the study groups.

### 4. Discussion

Several risk factors associated with developing severe forms of COVID-19 have been described, aging and the male sex being the most reported [9,10]. In the present study, the non-survivor patients were mainly male and had a median age of 63. The non-survivors presented a higher frequency of comorbidities, among which obesity and type 2 diabetes mellitus (T2DM) stood out; however, no significant differences were found when compared with the survivors. Chronic respiratory and ischemic heart disease were more frequent among non-survivors compared to survivors.

As the reports on other populations [11], in this study, the symptoms presented by patients with COVID-19 were not exclusively respiratory. In this regard, anosmia has been considered an initial neurological symptom in SARS-CoV-2 infection [12]. In our study, it was more frequent in the survivors' group. The respiratory symptoms that prevailed in the survivors were cough and dyspnea.

Additionally, non-survivors had more days of hospital stay, who had a median of 20 days, compared to those survivors, who had a median of 19 days. It has been reported that IMV use in patients with severe ARDS is related to the presentation of complications such as alveolar injury, which prolongs their hospital stay [13]. Regarding the outcome, the non-survivors had a higher frequency of requirements of IMV (93.0%) compared to survivors (60.9%). Most studies suggest that patients die due to the presentation of multifactorial systemic complications [14]. The group of non-survivors had lower $PaO_2/FiO_2$ at hospital admission compared to survivors.

The susceptibility to developing viral infections has been associated with the presence of genetic variants, and these are located mainly in genes that code for proteins that participate in the immune response [15]. Previous reports reveal that individuals carrying the risk alleles of *SERPINA1* (rs17580 (A) and rs28929474 (T)) have low levels of circulating A1AT, increasing the risk of SARS-CoV-2 infection in the Caucasian population [16]; however, in severe COVID-19, increased levels of A1AT have been observed along with other pro-inflammatory proteins, such as IL-10 and IL-6, suggesting that in severe COVID-19 cases, A1AT contributes to the appearance of a different inflammatory phenotype that could be associated with alterations in immunometabolism [17].

Through studies in vitro, A1AT has been observed to inhibit the entry of SARS-CoV-2 into the cell by binding to and inactivating transmembrane serine protease type 2 (TM-PRSS2) [18], preventing the union of the spike protein (S) of SARS-CoV-2 with the angiotensin-converting enzyme (ACE2) [19]. In the Mexican mestizo population we studied, both risk variants were found in low frequency, and there was no association between any of these polymorphisms and COVID-19 severity.

On the other hand, the presence of the T allele of rs35705950 (*MUC5B*) in patients with pre-existing idiopathic pulmonary fibrosis (IPF) seems to confer significant protection against severe COVID-19 (OR = 0.78, 95% CI 0.66–0.90) [20]. In the European population, having the T allele confers protection against the development of severe COVID-19 (OR = 0.75, CI95% 0.67–0.85) [21]. In the present study, there was no statistically significant association between the groups at the levels of genotypes and alleles. It should be noted that, unlike other reports, the patients included did not have a previous diagnosis of IPF.

At the protein level, those individuals carrying the T allele (in one or two copies) had higher levels of mucin 5B and survived COVID-19. The presence of mucin 5B is required for mucociliary clearance in the respiratory tract. This mechanism can remove pathogens and particles trapped in mucus, preventing them from accumulating in the upper and lower respiratory tract [22]. Mucin 5B levels have previously been reported to be decreased in the airway epithelium of patients with COVID-19 compared to uninfected individuals [23]. Our findings contribute to the fact that those patients with an accumulation of mucus in the airway require immediate attention to avoid complications.

The low frequencies of the minor allele of the variants in *SERPINA1* in the Mexican population made it difficult to find an association with the disease. However, it is relevant to study these variants due to their relationship with the development of chronic respiratory diseases. However, we do not report a significant difference for the rs37705950 in *MUC5B*. It is essential to highlight that this is the first study where this variant is analyzed in patients without a previous diagnosis of IPF, in addition to being the first study where plasma concentrations of mucin 5B are reported for patients with severe COVID-19 who were carriers of the allele risk (rs35705950/T). We carried out an analysis with the primary aim of evaluating the concentrations of the mucin 5B protein according to the outcome of the COVID-19. In addition, it would be important to include a healthy control group to evaluate genetic susceptibility and the concentrations of the mucin 5B in our population.

## 5. Conclusions

The polymorphisms rs28929474 and rs17580 of *SERPINA1* and rs35705950 of *MUC5B* are not associated with the risk of severe COVID-19 in the Mexican population. COVID-19 survivors patients bearing one or two copies of the rs35705950 risk allele have higher plasma levels of mucin 5B.

**Supplementary Materials:** The following supporting information can be downloaded at: https://www.mdpi.com/article/10.3390/cimb44080226/s1. Figure S1: The LD of rs17580 and rs28929474 of the *SERPINA1* gene in patients with COVID-19 survivors and non-survivors. We show $r^2$ (by Haploview v.4.1). Figure S2: Concentrations of mucin 5B in plasma from patients with COVID-19. Table S1. Genotype and allele frequencies of *SERPINA1* and *MUC5B* polymorphisms in patients included in the present study. Table S2. Genotype and allele frequencies of *SERPINA1* and *MUC5B* polymorphisms in patients with COVID-19 and ARDS classified by $PaO_2/FiO_2$.

**Author Contributions:** Conceptualization, R.F.-V., I.F.-G. and G.P.-R.; methodology, S.G.-C. and A.V.-A.; software, J.A.-P. and D.V.-P.R.; validation, L.C.-G. and A.M.-M.; formal analysis, I.B.-R. and R.d.J.H.-Z.; investigation, S.G.-C., R.F.-V. and G.P.-R.; resources, K.J.N.-Q., M.M. and I.P.G.-G.; data curation, A.C., I.A.G.-P. and O.Z.-G.; writing—original draft preparation S.G.-C., A.V.-A., I.B.-R. and L.C.-G.; writing—review and editing, R.F.-V. and G.P.-R.; visualization, R.d.J.H.-Z., K.J.N.-Q., M.M. and I.P.G.-G.; project administration, S.G.-C. and G.P.-R.; funding acquisition, R.F.-V. All authors have read and agreed to the published version of the manuscript.

**Funding:** This work is supported by the allocated budget to research (R.F.-V., HLA Laboratory) from the Instituto Nacional de Enfermedades Respiratorias Ismael Cosío Villegas (INER).

**Institutional Review Board Statement:** The study was conducted according to the guidelines of the Declaration of Helsinki and approved by the INER committee on bioethics and research (protocol number C53–20).

**Informed Consent Statement:** Informed consent was obtained from all subjects involved in the study.

**Data Availability Statement:** The datasets generated for this study can be found with the ClinVar accession numbers SCV002546355 and SCV002546357.

**Conflicts of Interest:** The authors declare no conflict of interest.

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
