# Peer review of "COVID-19 Survivor Patients Carrying the Rs35705950 Risk Allele in MUC5B Have Higher Plasma Levels of Mucin 5B"

_cimb, doi:10.3390/cimb44080226_

Round 1
Reviewer 1 Report
In the manuscript entitled “COVID-19 survivor patients carrying the rs35705950 risk allele in MUC5B have higher plasma levels of Mucin 5B”, the authors investigated the genetic polymorphisms of SERPINA1 and MUC5B in patients diagnosed with COVID-19. They aimed to identify polymorphisms associated with severe COVID-19. In addition, they measured the plasma Mucin 5B. Among the 1,258 patients diagnosed with COVID-19, less than 5% were identified with the risk alleles of the three polymorphisms, suggesting that polymorphisms rs28929474 and rs17580 of SERPINA1 and rs35705950 of MUC5B are not associated with the risk of severe COVID-19. On the other hand, the survivor group with one or two copies of the rs35705950 risk allele has higher plasma levels of Mucin 5B. Generally speaking, the study is well-designed and the authors’ conclusions are supported by their convincing results. I believe this work can pave the road for COVID-19-related research. The only concern is that non-survivor patients were older than survivors. Although it may not affect the conclusion, it would be better to discuss it.
Author Response
Thank you for the observation. A linear regression analysis was performed to assess whether the significant differences in Mucin 5B concentrations depending on the outcome are not affected by possible confounding variables (age, sex, BMI). After this analysis, the significance value was preserved (p =0.0058) (lines 168-171).
Reviewer 2 Report
With interest, I read the manuscript cimb-1804639.
It is not a sin for a manuscript to be based on a quite negative study. However, I have several resevations.
Specific comments:
1. Please, provide the details on the HWE analysis, i.e. the results (significances).
2. Please, add the significances to Tables 2, 1S, 2S.
3. Please, calculate the ORs with 95% CIs for all analyzed models (from Tables 2, 1S, 2S) and show them in the new tables/suppl. tables.
4. Please, do not present any data in the Discussion.
5. Lines 107-109: “The Kolmogorov-Smirnov test was performed to evaluate the distribution of the quantitative variables, and we applied parametric or non-parametric tests as appropriate.”. Which exactly? From Table 1 it seem that Mann-Whitney U test only, i.e. what was a parametric test?
6. Lines 114-115: “We employed the Wilcoxon-Mann-Whitney test to compare the protein level in the study groups.”. Because the distribution was different than normal? You mean Mann-Whitney-U test, more precisely, i.e. unpaired test?
7. Lack of the healthy control group, at least small is a limitation. Please, mention.
8. Line 239. What is “Mucin 5B.6.”?
9. Figure 1 is very raw. Please, elaborate. The legend, too. What is shown (exactly)? Suppl. figure 2 as well; what is “MUCB5$Outcome” btw.?
10. What happens if you adjust the results of paragraph “3.4. Evaluation of Mucin 5B plasma levels” for age, sex, BMI? Please, provide.
11. Line 167: “(0.0 vs. 0.17 ng/mL)”. IQRs?
12. What you call haplotype analysis is in fact a pairwise LD analysis (of course it would correspond to haplotype analysis but it is not exactly the same). Please, correct throughout the manuscript, including Suppl. Figure 1. By the way, you do not write what you show – please correct. The best if you gave both r2 and D’ in two separate plots.
Author Response
Thank you for the observation. We attachment file.

Round 2
Reviewer 2 Report
The Author have sufficiently addressed my comments.